# Druggable Metabolic Vulnerabilities Are Exposed and Masked during Progression to Castration Resistant Prostate Cancer

**DOI:** 10.3390/biom12111590

**Published:** 2022-10-28

**Authors:** Stephen Y. C. Choi, Caroline Fidalgo Ribeiro, Yuzhuo Wang, Massimo Loda, Stephen R. Plymate, Takuma Uo

**Affiliations:** 1Vancouver Prostate Centre, Vancouver, BC V6H 3Z6, Canada; 2Department of Urologic Sciences, Faculty of Medicine, University of British Columbia, Vancouver, BC V5Z 1M9, Canada; 3Department of Experimental Therapeutics, BC Cancer Agency, Vancouver, BC V5Z 1L3, Canada; 4Department of Pathology and Laboratory Medicine, Weill Cornell Medicine, New York-Presbyterian Hospital, New York, NY 10021, USA; 5New York Genome Center, New York, NY 10013, USA; 6Division of Gerontology and Geriatric Medicine, Department of Medicine, University of Washington, 850 Republican St., Seattle, WA 98109, USA; 7Geriatrics Research Education and Clinical Center, VA Puget Sound Health Care System, Seattle, WA 98108, USA

**Keywords:** androgen receptor, cancer metabolism, prostate cancer, Warburg’s effect, lactate, fatty acids, drug resistance

## Abstract

There is an urgent need for exploring new actionable targets other than androgen receptor to improve outcome from lethal castration-resistant prostate cancer. Tumor metabolism has reemerged as a hallmark of cancer that drives and supports oncogenesis. In this regard, it is important to understand the relationship between distinctive metabolic features, androgen receptor signaling, genetic drivers in prostate cancer, and the tumor microenvironment (symbiotic and competitive metabolic interactions) to identify metabolic vulnerabilities. We explore the links between metabolism and gene regulation, and thus the unique metabolic signatures that define the malignant phenotypes at given stages of prostate tumor progression. We also provide an overview of current metabolism-based pharmacological strategies to be developed or repurposed for metabolism-based therapeutics for castration-resistant prostate cancer.

## 1. Introduction

Prostate cancer (PCa) is the second most common cancer and the second leading cause of cancer death in American men. In 2022, 268,490 new cases will be diagnosed, and there will be an associated 34,500 deaths [1]. The major challenge in PCa therapy is to improve the outcome from lethal castration-resistant prostate cancer (CRPC), which almost inevitably emerges following anti-androgen therapy of advanced PCa [2,3,4,5]. Progression to CRPC and resistance to androgen receptor (AR) targeting therapies such as androgen synthesis inhibitors (abiraterone) and AR-ligand inhibitors (enzalutamide, apalutamide, and darolutamide) is often associated with persistent AR signaling due to multiple mechanisms such as increased AR copy number, activating point mutations, and constitutively active splice variants (AR-Vs) [3,5,6,7,8]. Tumor plasticity driven by continual stepwise targeting of AR has garnered recent attention and may lead to AR negative cells driven by fibroblast growth factor (FGF) or neuroendocrine transdifferentiation mechanisms [9,10,11,12]. The AR domains outside of the ligand-binding domain are attractive sties for pharmacological interference [12,13,14,15,16]. However, the compound EPI-7386 targeting the AR-N-terminal transactivation domain is still in Phase 1 trial, and degrading agents proteolysis targeting chimeras (PROTACs) against AR-Vs are still in preclinical development [17,18,19,20]. Much effort has been made to therapeutically target PCa-specific cell surface receptors such as prostate-specific membrane antigen (PSMA) [21,22,23,24], DNA damage and repair (DDR) pathways [25,26], and bone metastasis niches [27,28]. However, results from clinical trials indicate that these treatments continue to have issues including but not limited to the targeting of specific patient populations and resistance mechanisms, and tumor heterogeneity [29,30].

One hundred years have passed since Otto Warburg’s pioneering work to report aerobic glycolysis in tumors [31,32,33]. Nevertheless, antiglycolytic agents such as 2-deoxyglucose (2-DG) had never been clinically translated [34,35]. On the other hand, Sidney Farber successfully introduced antifolic agents in 1948 [36], which became the basis for many of the most widely used chemotherapies [37]. Early studies focused on exploiting cancer metabolism proved to be an effective strategy for some cancers. However, serial discoveries of genetic drivers such as tumor suppressors and oncogenes moved the focus of therapeutic strategies away from metabolism, which was often thought to be a mere passenger event [35,38,39,40]. The situation drastically changed when the link between oncogenic drivers and metabolic reprogramming began to be appreciated in the early 2000s [39,40,41,42,43]. Altered metabolism not only supports and maintains tumor phenotypes by meeting their energetic and metabolic demands, it also allows for further progression to metastases [44,45]. Metabolic reprogramming is now firmly established as a hallmark of cancer and is attractive because it represents unique vulnerabilities that can be therapeutically targeted [39,41]. Indeed, promising and innovative therapeutic strategies are emerging owing to the resurgent appreciation of cancer metabolism [35,46]. Their antitumor effects may simply reflect the promotion of energy consumption and subsequent starvation-induced cell death, or, more ideally, be related to active tumoricidal activities [42].

There is an urgent need for exploring new actionable targets other than AR to improve outcome from lethal CRPC [47,48,49]. In this regard, it is important to understand the relationship between distinctive metabolic features, AR signaling, genetic drivers in PCa, and the tumor microenvironment (TME) to identify metabolic vulnerabilities [50,51,52,53,54]. A one-size-fits-all approach has not been successful in anti-CRPC therapy [5,8,48]. Therefore, in this article, we will discuss whether therapeutic targeting of tumor metabolism can arise as precision medicine to give patients a “tailor-made” therapy based on their metabolic phenotypes [55]. To this end, we will provide an overview of the current knowledge of cell autonomous metabolic alterations and metabolic interactions in the TME during tumor initiation and progression, and introduce potential therapeutic targets that emerge from unique metabolic perspectives in PCa.

## 2. Metabolic Reprogramming during Progression to CRPC

### 2.1. Zinc-Driven Metabolism

In benign prostate epithelial cells, the zinc transporter ZIP1 facilitates the intracellular accumulation of zinc ions. This leads to the inhibition of m-aconitase (ACO2) in mitochondria and truncates the tricarboxylic acid (TCA) cycle to accumulate and release citrate into the prostatic fluid [56,57]. Androgen signaling is critical in supporting continuous citrate production. Because of the increase in zinc, the citrate synthase substrate oxaloacetate (OAA) is not derived from the TCA cycle but is generated by the action of mitochondrial aspartate aminotransferase (GOT2) on aspartate, which in turn is imported into the cells through the excitatory amino acid transporter EAAC1 (SLC1A1) [58,59]. Another substrate in mitochondria, acetyl-CoA, is supplied in association with an increased expression of pyruvate dehydrogenase E1 component subunit alpha (PDHE1α) [60]. Depletion of ZIP1 represents an essential early event in the development of PCa malignancy, whereby lower intracellular zinc levels relieve ACO2 to establish a complete TCA cycle [61] (Figure 1). Contrary to other types of cancer cells, malignant transformation in PCa involves the conversion from “aerobic glycolytic” benign cells to “oxidative” cancerous cells [50,51,62]. These bioenergetic alterations are reflected by the general low avidity of ^18^F-fluorodeoxyglucose (FDG) in primary PCa [50,51,62,63,64].

### 2.2. AR Alterations

Localized PCa is treated primarily by radical prostatectomy or radiotherapy, while recurrent disease requires androgen deprivation therapy (ADT). PCa progression towards late-stage disease is primarily driven by alterations in the AR signaling pathway (Figure 1), but tumors will inevitably bypass androgen dependence and develop resistance to ADT. Metastatic CRPC (mCRPC) was shown to have multiple genomic alterations related to the AR, including gene amplification and mutations [65,66]. The first leads to increased levels of AR expression [67], while mutated AR can show gain-of-function and enhanced activity [65]. Certain AR mutations can also affect receptor specificity, allowing other steroid hormones such as estrogen, progesterone and glucocorticoids to bind to AR and induce transcription of its target genes [68]. Proteins that interact with AR, such as chromatin remodelers and cofactors, also present genomic aberrations [69], confirming the pivotal role of the AR pathway in prostate carcinogenesis. Alternative RNA splicing is also observed, which gives rise to constitutively active AR-variants (AR-V) [70], of which AR-V7 is considered the most critical driver of resistance to ADT [71].

Second-generation AR antagonists such as enzalutamide [72] and apalutamide [73] can impair ligand-dependent AR-mediated transcription and helped improve survival of mCRPC patients [74,75]; ultimately, however, patients will progress to lethal disease. Neuroendocrine prostate cancer (NEPC) is mainly characterized by the lack of expression of both AR and AR downstream targets, and by the expression of neuroendocrine differentiation markers. It can develop de novo or, more prevalently, can be induced by AR-mediated therapy as treatment-emergent NEPC (t-NEPC) [76,77]. Molecularly, these types of PCa are marked by *TP53* and *RB1* loss, and by *AURKA* and *MYCN* amplification [78]. The emergence of tumors that lack both AR and neuroendocrine markers, known as double-negative PCa (DNPC) [11], is also related to anti-AR therapies. Regardless of the mechanism by which cells can achieve androgen independence, they represent a therapeutic hurdle that requires alternative strategies to improve patient survival. Platinum-based chemotherapy is commonly used for these patients, and several clinical trials have investigated the efficacy of combination therapies, such as docetaxel and cisplatin [79], and etoposide and carboplatin [80]; however, these studies report unfavorable toxicity profiles. Other studies have shown promising data when targeting alternative signaling pathways in advanced, androgen-indifferent settings: FGF/mitogen-activated protein kinase (MAPK) blockade was shown to be effective in impairing tumor growth in preclinical models of DNPC [11]. Delta-like protein 3 (DLL3)-directed therapy showed promising results in NE-like patient-derived xenograft (PDX) models [81]. Inhibition of both Aurora kinase A (AURKA) and poly (ADP-ribose) polymerase (PARP) in preclinical models of DDR-defective NEPC was also able to suppress tumor growth [82].

A better understanding of the cell biology in advanced PCa settings can help the development of newer and directed therapies; particularly, metabolic adaptations during PCa progression can be exploited. Hanahan and Weinberg defined the hallmarks of cancer as acquired cellular alterations that can sustain growth and allow cells to proliferate [38,39]. Six physiologic changes that are fundamental for tumorigenesis were initially established: self-sufficiency in growth signals, insensitivity to growth-inhibitory (antigrowth) signals, evasion of programmed cell death (apoptosis), limitless replicative potential, sustained angiogenesis, and tissue invasion and metastasis. Recently, the concept has been expanded to eight hallmarks that include: sustaining proliferative signaling, evading growth suppressors, resisting cell death, enabling replicative immortality, inducing/accessing vasculature, activating invasion and metastasis, reprogramming cellular metabolism, and avoiding immune destruction [41]. The inclusion of metabolic reprograming as a core hallmark of cancer reflects the abundant literature in the field that has shown the key role of metabolic adaptations in tumor growth. AR is known to play a role in tumor cell metabolic reprogramming during PCa progression. For example, metabolic signatures associated with AR-V7 have shown a decrease in citrate levels with a concomitant increase in glycolysis and dependence on the reductive carboxylation of glutamine [83]. Sterol response element-binding proteins (SREBPs), the transcription factors controlling the expression of lipid metabolism genes, are also androgen regulated [84]. Up-regulation of SREBPs and lipogenic genes contributes to CRPC progression [85]. Fatty acid synthase (FASN), the principal enzyme in de novo lipogenesis and an SREBP target gene, is overexpressed in CRPC [86,87]. In fact, PCa displays differential metabolic phenotypes depending on the stages of tumor progression [50,51]. Tumor imaging with positron emission tomography (PET) radiotracers helps determine what bioenergetic phenotype dominates in PCa tumors (glycolytic, lipogenic, or oxidative) [88,89]. Tumor uptake of ^11^C-acetate is indicative of a lipogenic phenotype, while high avidity to ^18^F-FDG is supportive of a glycolytic gene signature in advanced adeno-PCa, NEPC and DNPC [90,91,92,93,94,95,96]. Mitochondrial membrane potential (*ΔΨm*) drives uptake of the agent 4-[^18^F]fluorobenzyl triphenylphosphonium (^18^FBnTP), which reflects oxidative tumors [97]. Next, we will discuss the major metabolic alterations observed throughout PCa progression.

## 3. Major Bioenergetic Sources

Due to their highly proliferative behavior, tumor cells need to rewire their metabolism to sustain a higher energetic demand [42,46,98]. The major examples are the altered utilization of glucose and endogenous and exogenous fatty acids (FAs). In addition, several other nutrients such as glutamine, acetate, and fructose are used by cancer cells to fuel their growth.

### 3.1. Glucose Metabolism

The quintessential example of altered cancer metabolism involves the aberrant utilization of glucose for increased lactic acid production. As a phenomenon first observed a century ago by Otto Warburg [99,100,101], it is traditionally described as an overreliance on the glycolysis pathway even in the presence of abundant oxygen (termed “aerobic glycolysis” or the “Warburg effect”). Rather than fully catabolizing glucose into carbon dioxide through glycolysis in conjunction with the TCA cycle and oxidative phosphorylation (OXPHOS), many cancer cells truncate this process at the end of glycolysis, diverting pyruvate towards lactic acid production instead. This is evidenced by the in vivo conversion of [1-^13^C] pyruvate to [1-^13^C] lactate based on hyperpolarized ^13^C magnetic resonance spectroscopic imaging (MRSI) [102]. The accompanied increase in glucose consumption and lactic acid secretion is widely observed across multiple cancer types. More significantly, this metabolic reprogramming is commonly associated with poor prognosis and increased aggressiveness in an extensive spectrum of cancers, including breast, lung, prostate, kidney, head and neck, colorectal, and liver cancer, to name a few [103,104,105,106,107].

Altered glucose metabolism is unique in the PCa context because, as discussed above, normal prostate epithelium has a distinct baseline metabolic profile. Due to the tissue-specific truncation of the TCA cycle for citrate production, the metabolic alteration that signals prostate carcinogenesis is, paradoxically, a reversion to “normal” glucose metabolism (transition from energy inefficient glycolytic metabolism to energy efficient oxidative metabolism) [108]. Nevertheless, elevated glucose consumption and lactic acid secretion remains a relevant metabolic alteration as the disease progresses towards more treatment-resistant and aggressive phenotypes [50]. This is exemplified by the fact that, while FDG-PET is not a useful diagnostic tool for imagining early PCa, it regains utility for the assessment of metastatic nodules and for the more aggressive NEPC [93,95]. As such, altered glucose metabolism is an important characteristic associated with PCa development and progression, even if it is slightly more nuanced than traditionally conceived.

Since glucose remains a key energy source in PCa metabolism, whether as fuel for adenosine triphosphate (ATP) production or as a precursor for biomolecule synthesis, multiple components of its import and catabolism can be deregulated as PCa progresses towards more advanced stages (Figure 2). AR signaling governs PCa metabolism by regulating components in the glycolytic pathway (glucose transporter GLUT1, hexokinase HK1, HK2, and 6-phosphofructo-2-kinase/fructose-2,6-bisphosphatase PFK2/PFKFB) and in pyruvate flux into mitochondria (PDH and mitochondrial pyruvate carrier MPC2). At the top of the glycolytic pathway is the cell’s ability to import glucose from the external environment. Expression of glucose transporters can become elevated to facilitate increased glycolytic flux. For example, GLUT1 (SLC2A1) is the primary glucose transporter implicated in tumorigenic transformation and is frequently considered a general therapeutic target in many advanced cancers [109,110,111]. Various studies have demonstrated that GLUT1 expression is correlated with poor prognosis and adverse outcomes in PCa [90,112]. Functionally, increased expression of GLUT1 can promote a more aggressive and glycolytic phenotype in PCa, especially in response to increased AR signaling [50,113,114,115]. While GLUT1 remains the most-studied glucose transporter, other family members such as GLUT3 (SLC2A3) and GLUT4 (SLC2A4) may also play a role in facilitating increased glucose uptake in advanced, treatment-resistant PCa [114,116,117,118].

Many other components of the glucose metabolic pathway are tightly associated with increased glucose uptake and are often assessed together as part of an overall glycolytic profile [90,119,120,121]. The immediate next steps following glucose uptake are considered critical regulatory junctions that determine glucose commitment towards glycolysis. Unsurprisingly, increased expression and altered functionalities of the relevant enzymes have been reported in the PCa context. For example, both the hyperactivation and mitochondrial localization of HK2, which supports Warburg-type tumors, facilitate PCa tumorigenesis and aggressiveness. This is especially prominent in a phosphatase and tensin homologue (PTEN)-deficient context, which results in constitutive activation of phosphatidyl inositol 3-kinase (PI3K)/AKT signaling [122,123,124]. The fact that HK2 functions as a regulatory nexus and is a downstream target of PI3K/AKT signaling makes it an attractive therapeutic target for CRPC [125,126,127]. Another critical component of the glycolysis pathway, phosphofructokinase (PFK), is also upregulated during PCa progression. The rate-limiting glycolytic enzyme PFK1 converts fructose-6-phosphate to fructose-1,6-bisphosphate (1,6-FBP), which is then converted to glyceraldehyde-3-phosphate and dihydroxyacetone-3-phosphate [128,129]. PFK1 is allosterically regulated by the negative regulator citrate and the positive regulator 2,6-fructose bisphosphate (2,6-FBP). 2,6-FBP is produced by the PFK2 family of enzymes, which control intracellular 2,6-FBP levels through their dual kinase and phosphatase activities [128,129]. Increases in PFK1 and PFK2/PFKFB2 can all contribute to higher rates of glucose utilization and greater cancer cell survival in advanced PCa [130,131]. Conversely, activation of PFKBP4 leads to PFK1 inhibition, which directs glucose 6-phosphate (G6P) from glycolysis to the pentose phosphate pathway (PPP) for generation of nicotinamide adenine dinucleotide phosphate (NADPH) and nucleotide precursors. Relatedly, other glycolytic enzymes in the ten-step pathway can also help maintain increased glycolytic flux, with increases in aldolase expression being one contributor to PCa proliferation [132].

The end of the glycolysis pathway represents another major regulatory junction in glucose metabolism. Pyruvate is a central hub, integrating cellular energetics from different metabolic compartments. Many important enzymes regulate the final conversion of phosphoenolpyruvate into pyruvate and determine its ultimate trajectory either towards lactic acid production or into mitochondria for the TCA cycle. Pyruvate kinase is one such enzyme that plays a critical regulatory role in PCa progression, depending on the predominant isoform present in the cell. It has a liver and red blood cell isozyme (PKLR) and a muscle isozyme with two alternatively spliced forms, PKM1 and PKM2 [133]. PKM1 is a constitutively active tetrameric enzyme, while the allosteric binding of 1,6-FBP to PKM2 promotes its tetrameric formation from the less active dimer. An abundance of PKM1 suppresses PCa development by promoting more complete glucose catabolism, thereby limiting glucose entry into the PPP and reducing the production of nucleotides necessary for proliferation [134]. Alternatively, PKM2 functions as a more typical oncogene, and its overexpression is implicated in multiple cancer types [135,136]. Metabolically, PKM2 has reduced enzymatic activity and slows down pyruvate production, allowing more upstream biomolecules to be available for other biosynthetic processes [137]. A recent report suggests that deoxyribonucleic acid (DNA)-dependent protein kinase (DNA-PK) may play a role in regulating increased glycolytic flux by phosphorylating aldolase and PKM2 to increase their enzymatic activities [138]. Interestingly, PKM2 also possess non-metabolic kinase activity and can function as a transcription factor, thereby promoting metastasis and regulating responses to hypoxia in PCa [139,140,141,142,143].

Glycolysis persists with pyruvate reduction to lactate in the cytosol. Alternatively, pyruvate flux into mitochondria is the first step of TCA-driven glucose oxidation and is mediated by MPC, which is a hetero-oligomer composed of MPC1 and MPC2 [144,145,146]. Changes to regulators that determine pyruvate’s metabolic fate also contribute significantly to PCa progression. AR positively regulates MPC activity by increasing expression of MPC2 to support mitochondrial pyruvate oxidation in AR-driven PCa [147]. MPC inhibition resulted in activating transcription factor 4 (ATF4)-mediated activation of “integrated stress response” and delayed cell cycle progression [147]. Furthermore, it reduces AR-dependent PCa growth by decreasing the amount of pyruvate entering mitochondria, thereby restricting important TCA outputs such as acetyl-CoA for lipogenesis and other intermediates for OXPHOS [147]. Pharmacological inhibition of MPC also promotes metabolic alterations that rely on differential mitochondrial substrate utilizations, including using glutamine and branched chain amino acids for anaplerotic reactions to replenish TCA cycle intermediates and support OXPHOS [146,148,149]. Indeed, simultaneous inhibition of MPC and anaplerotic pathways exhibits drastic antitumor activities [148,149]. Intriguingly, reductions in MPC expression can increase PCa aggressiveness by promoting lineage reprogramming into AR-independent NEPC [150]. Moreover, MPC overexpression reverses NEPC differentiation and helps adeno-PCa regain sensitivity to enzalutamide [150]. Beyond MPCs, the metabolic switch between glycolysis and mitochondrial oxidation is facilitated by the pyruvate dehydrogenase complex, its inhibitory phosphorylation by associated kinases, and opposing phosphatases (i.e., pyruvate dehydrogenase kinases [PDKs] and pyruvate dehydrogenase phosphatase catalytic subunit 1 [PDP1]) [151]. They are implicated during PCa progression and play a critical role in modulating acetyl-CoA production [152,153,154,155]. Increased PDHA1 and PDP1 activities have been shown to enhance PDK activity, lipogenesis, and glutamine dependence in PCa [156,157]. Similarly, PDK1 can regulate PCa progression and local invasion [158,159], while increased expression of PDK4 is correlated with poor prognosis in PCa [160].

Lactate dehydrogenases (LDHs) catalyze pyruvate conversion into lactic acid with regeneration of nicotinamide adenine dinucleotide (NAD^+^). LDHA has a higher affinity for lactate production from pyruvate, while LDHB possesses a higher affinity for lactate reconversion to pyruvate. Multiple reports have indicated that increased LDHA expression can confer treatment resistance and helps PCa adapt a more glycolytic and aggressive phenotype [161,162,163,164]. Furthermore, fibroblast growth factor receptor 1 (FGFR1) can mediate the balance between LDH isoforms, increasing the stability of LDHA and reducing the expression of LDHB, thereby tipping the metabolic flux in favor of increased lactic acid production [165]. Finally, increased lactic acid production also engenders increased lactic acid efflux in PCa cells. Changes to monocarboxylate transporter (MCT) expression help facilitate the final excretion of lactate into the TME. This primarily involves the increased expression of the proton symporter MCT4 (SLC16A3), which has been shown to promote CRPC and NEPC aggressiveness and suppress anticancer immunity [105,121]. Alternatively, co-expression of MCT1 (SLC16A1) and MCT4, especially MCT1 in PCa cells and MCT4 in the stromal compartment, also contribute to a more aggressive PCa phenotype [166,167]. It is well established that increased lactic acid secretion by cancer cells plays a critically important role in promoting many downstream cancer hallmark characteristics, including angiogenesis, metastasis, resistance to hypoxia and weak-base therapeutics, and the suppression of anticancer immunity [168,169].

### 3.2. Fructose

Fructose plays an important metabolic role in prostate cells [170]. Seminal fluid is rich in fructose, which is endogenously generated from glucose by the sequential actions of sorbitol dehydrogenase and aldose reductase [171]. The former is an AR-driven gene that is expressed in the seminal vesicles and in malignant PCa [172]. Moreover, emerging evidence suggests that dietary fructose or seminal fructose can be incorporated for energy and integrated into the glycolytic pathway by highly expressing the main fructose transporter GLUT5 (SLC2A5), suggesting that PCa possibly utilizes hexoses other than glucose [173,174]. This is another explanation for why clinical applicability of FDG-PET is limited in PCa diagnostics [114,170].

### 3.3. Amino Acid Metabolism

While glucose metabolism is considered the typical pathway for maintaining cellular energetic requirements, alterations to amino acid utilization also contribute significantly to PCa development. These changes generally facilitate anabolic processes, and they play a more fundamental role in producing biomolecule precursors for proliferation than in generating “energy” per se (for example, increased nitrogen catabolism for pyrimidine synthesis) [175]. Changes to the metabolism of many amino acids have been observed in PCa, ranging from glutamine and glycine to phenylalanine and sarcosine [176]. Indeed, when considering altered cancer metabolism more broadly beyond glucose, changes to the metabolic landscape may involve multiple interrelated pathways that collectively contribute to disease progression and aggressiveness [52]. A selection of the more prominent amino acid pathways is reviewed here.

#### 3.3.1. Glutamine Metabolism

Glutaminolysis is another key pillar of altered cancer metabolism. Most practically, this phenomenon is understood as the need to supplement glutamine into cell culture media to cultivate in vitro cancer cell growth [175,177]. On a more mechanistic level, it is traditionally described as the import of glutamine that is then converted into glutamate, which is further metabolized into alpha-ketoglutarate (α-KG) as it enters the TCA cycle. This pathway can help replenish TCA cycle intermediates, especially if OXPHOS is to be maintained. Furthermore, various anabolic processes can be fueled by this pathway as metabolic intermediates exit the TCA cycle. Malate can be converted into pyruvate for lactic acid production or gluconeogenesis. OAA can be converted into aspartate for nucleotide synthesis. Citrate and acetyl-CoA can also be redirected towards FA production [178]. Similar to other pathways involved in PCa development, alterations to glutamine metabolism can also be androgen regulated [179,180].

The increased utilization of glutamine is facilitated by increased expressions of amino acid transporters such as alanine serine cysteine transporter 2 (ASCT2; SLC1A5) and L-type amino acid transporter 1 (LAT1; SLC7A5). Both are associated with PCa progression and treatment resistance [181,182,183,184]. Downstream of glutamine transport are two forms of glutaminase (GLS) encoded by distinct genes, designated GLS1 and GLS2. GLS1 has two splicing isoforms, kidney-type glutaminase (KGA) and the more active isoform glutaminase C (GAC) [185]. Increases in amino acid transporter expressions often coincide with increases to these other glutaminolytic enzymes [157,175,186,187,188,189]. More specifically, the overexpression of GLS for the first-step conversion of glutamine to glutamate in PCa cells can be MYC-driven and is associated with the maintenance of a cellular redox state that confers radioresistance [190]. GLS1 overexpression can also more generically promote PCa cell growth and facilitate metastasis [191,192]. Alternatively, resistance to ADT is also reportedly facilitated by a switch to the androgen-independent isoform GAC [180]. Finally, either transaminase or glutamate dehydrogenase (GDH) converts glutamate into α-KG when PCa cells enter an energy-restricted state, especially if typical glycolysis becomes unavailable [157]. Mitochondrial catabolism of the branched chain amino acids leucine, isoleucine, and valine is also critical as a major source of cellular energy via generation of acetyl-CoA and succinyl-CoA [193].

In relation to glutamine metabolism, reliance on glutamine-derived carbon and precursor molecules for biosynthesis is also important in cancers that harbor mitochondrial defects or OXPHOS mutations. When typical TCA cycle becomes hampered by these deficiencies, glutaminolysis allows cancer cells to maintain the necessary pool of TCA cycle intermediates for various biosynthetic processes [194]. This can become especially pertinent in situations where altered electron transport chain (ETC) capacities factor into PCa disease progression [195]. Alternatively, altered glutamine metabolism can be associated with changes to proline biosynthesis, even if proline metabolism itself is a double-edged sword that both promotes and suppresses tumorigenesis [196,197,198]. In the PCa context, it has been shown that increased proline biosynthesis and reduced proline degradation is associated with reduced biochemical disease-free survival and can be regulated by C-MYC [189,199].

#### 3.3.2. One-Carbon Metabolism

Another prominent aspect of amino acid metabolism involves serine, glycine, and methionine in relation to the cellular availability of methyl groups and precursor molecules [200]. 3-Phosphoglycerate (3-PG) is a glycolytic intermediate and a precursor of serine biosynthesis. 3-PG is converted into serine by three sequential enzymatic reactions mediated by phosphoglycerate dehydrogenase (PHGDH), phosphoserine aminotransferase (PSAT1), and 1-3-phosphoserine phosphatase (PSPH) [200,201,202]. The mutual conversion of serine and glycine is coupled with the folate cycle. Folate metabolism involves a set of paralogous enzymes which catalyze similar reactions in mitochondria and in the cytoplasm [200,201]. Cytosolic serine hydroxymethyltransferase (SHMT1) acts on glycine to generate serine. The mitochondrial isozyme SHMT2 facilitates the synthesis of glycine from serine along with the interconversion of tetrahydrofolate (THF) and 5,10-methylene-THF. In the cytosol, a single trifunctional protein MTHFD1 contains 5,10-methylene-THF dehydrogenase, 5,10-methenyl-THF cyclohydrolase, and 10-formyl-THF synthetase activity. On the other hand, dehydrogenase and cyclohydrolase activity is attributed to the bifunctional protein MTHFD2 (MTHFD2L), while MTHFD1L is a 10-formylt-THF synthetase.

Exogenous serine and glycine both provide important building blocks for protein, nucleic acid, and lipids synthesis [203,204]. Many enzymes involved in one-carbon metabolism is AR- or MYC-regulated [205,206], and an overall upregulation in this metabolic pathway plays a crucial role in facilitating PCa progression [199]. Furthermore, ATF4 has been shown to selectively activate the mitochondrial components of one-carbon metabolism, leading to enhanced cell growth and survival [207]. Alternatively, increased serine biosynthesis and one-carbon metabolism can be caused by PKCλ/ι deficiency in NEPC in an mTORC1/ATF4-driven manner [208]. Finally, there is evidence suggesting that increased levels of the glycine-derivative sarcosine as synthesized by glycine N-methyltransferase (GNMT) can promote PCa invasion and aggressiveness [209]. These new studies help develop game-changing therapeutic strategies, even though we still take advantage of a 75-year-old invention in the form of antifolate therapies [35,210].

One-carbon metabolism can be further linked to cysteine and sulfur metabolism, cellular redox balance, and histone methylation [200,205]. More specifically, it is linked to the polyamine biosynthetic pathway through the S-adenosyl-methionine (SAM)-dependent transfer of methyl groups [211]. In PCa, polyamine metabolism is frequently dysregulated. AR drives the expression of two key enzymes, ornithine decarboxylase (ODC1) and the SAM-decarboxylase AMD1, thus elevating polyamine to levels necessary for transformation and tumor progression [212,213]. ODC1 catalyzes the conversion of ornithine to putrescine. The further reactions to generate spermidine and spermine require decarboxylated SAM (dcSAM), which is obtained from the decarboxylation of SAM by AMD1 [211]. mTORC1 phosphorylates and activates AMD1 to integrate growth factor signaling to polyamine metabolism [213]. Increased polyamine metabolism is sustained via the methionine salvage pathway by methylthioadenosine phosphorylase (MTAP) to recycle SAM [211,214]. Targeting this dependency on MTAP is an attractive translatable therapeutic approach.

### 3.4. Lipid Metabolism

FAs play an important role in cell proliferation as they are the building blocks for cell and organelle membrane synthesis [215,216] (Figure 3). FAs are also a vital source of energy and can modulate signaling pathways through post-translational modifications. FAs can be endogenously synthesized through de novo lipogenesis (DNL) or exogenously acquired from the environment [215,216]. FASN, the principal enzyme in DNL, is a homodimeric enzyme composed of seven catalytic domains (β-ketoacyl synthase, malonyl/acetyl transferase, dehydrase, enoyl reductase, β-ketoacyl reductase, acyl carrier protein, and thioesterase) that promote the synthesis of palmitate from acetyl-CoA and malonyl-CoA [217]. The FASN substrate acetyl-CoA is mainly generated upon cleavage of citrate, which is derived from the TCA cycle or from reductive carboxylation in the cytosol. Alternatively, acetyl-CoA is directly synthesized by the action of cytosolic acetyl-CoA synthetase 2 (ACSS2) on acetate [218,219]. Acetate uptake is increased concomitantly with elevated FASN activity and serves as a basis for powerful PET imaging [92,220]. FASN is the only eukaryotic enzyme that is able to produce palmitate in humans, and its expression is mostly restricted to adipocyte and liver cells. Other tissues have low to undetectable DNL rates [221,222]. Palmitate can be further elongated and desaturated to generate the diverse spectrum of saturated and monounsaturated FAs (SFA and MUFA) in mammalian cells. Palmitate and its derivatives can support membrane synthesis, be utilized for energy production to sustain increased proliferation, be stored as an energy reservoir, trigger signaling of major oncogenic pathways, and be used as substrates for oncogenic post-translational modifications [223]. FASN expression is upregulated in several types of cancer and correlates with poor prognosis, such as colorectal cancer [224], endometrial carcinoma [225], mantle cell lymphoma [226], B-cell non-Hodgkin lymphoma [227], multiple myeloma [228], ovarian cancer [229], breast cancer [230], and PCa [231,232].

The intracellular uptake of exogenous FAs can happen through a protein-mediated process involving FA transporters such as the cluster of differentiation 36 (CD36). CD36 is a glycoprotein responsible for the transport of long-chain FAs into the cells [233] and for cellular signal transduction [234,235]. It is expressed on the surface of several human cell types. CD36 expression and activity can be modulated by several ligands in response to fat supply. It was shown that a high-fat diet can upregulate CD36 transcription through hyper-O-GlcNAcylation and activation of the NF-κB pathway. It can also modify CD36 amino acid residues to promote increased FA uptake [236]. The peroxisome proliferator-activated receptor gamma (PPARγ) can also modulate CD36 expression levels [237]. Post-translational modification of CD36 through palmitoylation of its cytoplasmic tails is observed and can increased receptor activity [238,239]. As metabolic rewiring plays a role in the survival of tumor cells, different studies have implicated CD36 in carcinogenesis. Leukemic stem cells expressing CD36 can increase FA uptake from adipose tissue and fuel FA oxidation (FAO) to meet energetic demands for survival [240]. Increased FA uptake through upregulation of CD36 was observed in breast cancer resistant to anti-HER2 therapy [241]. In addition to providing FA as fuel for cell growth, CD36 was shown to support the formation of blood vessels in melanoma cancer cells by interaction with extracellular matrix components [242]. In PCa, both FA uptake and CD36 levels are upregulated in human malignant tissue, while genetic ablation of the receptor in Pten-deficient mouse models can impair tumor burden [243]. CD36 seems to also play a role in metastasis, as it is transcriptionally upregulated in lung metastatic CRPC [244]. The crosstalk between endogenous synthesis and exogenous uptake of lipids is of therapeutic relevance. It has been shown that tumor cells upregulate the rate of DNL to favor saturated membranes over polyunsaturated ones, as highly unsaturated FA acyl chains can render membranes more susceptible to oxidative damage [245]. Interestingly, DNL rates were observed to increase in the prostate of *Pten*^−/−^
*Cd36*^−/−^ mice as cells try to compensate for the FA deficiency caused by uptake blockade [243]. Enhanced antitumorigenic effects by inhibiting both DNL and CD36 activity has been observed in preclinical models of PCa [243,246].

#### FA Storage and Utilization

As cancer cells upregulated uptake and synthesis of FA, they must also modulate storage and utilization of lipids (Figure 3). FA can be stored in lipid droplets. Lipid droplets are intracellular organelles that consist of a phospholipid monolayer within which neutral lipids can be stored. This is a process mediated by budding from the endoplasmic reticulum (ER) membrane, and subsequent growth is facilitated by the transferal of triacylglycerol (TAG) and cholesterol esters [247]. Cancer cells can upregulate lipolysis to utilize stored FAs. TAG can be hydrolyzed to diacylglycerol (DAG) and free FAs by the action of adipose triglyceride lipase (ATGL). DAG can then be converted to monoacylglycerol (MAG) and free FAs by hormone-sensitive lipase (HSL). Cells can obtain glycerol and free FAs by monoacylglycerol lipase (MAGL) activity [248]. All these steps are crucial for tumor cells that have high energetic demands; MAGL, for example, was shown to be upregulated in both advanced cancer cells and primary tumors, increasing free FA levels and modulating migration and tumor growth [249].

In addition to lipolysis, cancer cells also rely on the oxidation of FAs. Oxidation of FAs is observed in both mitochondria and peroxisomes, and is a major metabolic energy source. While peroxisomal FAO does not generate energy per se, very long and branched chain FAs cannot be processed directly in the mitochondria and need to be initially broken down in peroxisomes [250]. Peroxisomal FAO is increased in PCa, with a higher expression level and enzymatic activity of D-bifunctional protein (DBP) [251]. α-Methylacyl-CoA racemase (AMACR), an enzyme involved in both mitochondria and peroxisome FAO, has also been reported to be increased in PCa [251,252]. Peroxisomes are a single-membrane bound organelle, with ABCD proteins importing the CoA esters of FAs [253]. On the other hand, mitochondrial FA import utilizes the carnitine shuttle, which is composed of carnitine palmitoyltransferases (CPT1, CPT2) and the carnitine–acylcarnitine translocase (SLC25A20). Subsequently, acetyl-CoA derived from FAO enters the TCA cycle, and NADH drives ATP synthesis through OXPHOS. The outer mitochondrial membrane (OMM) enzyme CPT1 is the rate limiting enzyme in FAO and is repressed by the FASN substrate malonyl-CoA. It controls the balance between FA synthesis and oxidation depending on metabolic demands [216].

## 4. Metabolites and Gene Regulation

As early as 1961, the link between metabolism and transcription has already been recognized in *Escherichia coli* as the “lac operon theory”. At the molecular level, the presence of lactose and the absence of glucose turns on the expression of genes involved in lactose metabolism [254,255]. Now, a growing body of evidence suggests that, in mammalian cells, specific metabolites mediate the crosstalk between cellular metabolism and gene regulation in multiple layers [256]. Cholesterol deprivation promotes the dissociation of precursor SREBP in complex with SREBP cleavage-activating protein (SCAP) from insulin-induced gene (INSIG1) at ER membranes, allowing it to move to the Golgi apparatus. There, it undergoes proteolytic cleavage to release the mature form of SREBPs, which act as transcription factors in nucleus [257]. MondoA and its binding partner Mlx are localized at the OMM [258,259]. Binding of the glycolysis intermediate G6P to MondoA initiates the translocation of the MondoA/Mlx complex to the nucleus, where it drives TXNIP and its paralog ARRDC4, which in turn downregulates the expression of glycolytic enzymes in a feedback manner. In this setting, elevated mitochondrial ATP levels regulate MondoA functions through the production of G6P by mitochondria-bound hexokinases [258,259]. Arginine is a conditionally essential amino acid, particularly in cells with low or null expression of argininosuccinate synthethase 1 (ASS1), which catalyzes arginine synthesis from citrulline and aspartate in concert with argininosuccinate lyase [260]. Arginine starvation activates the p38 MAPK pathway to promote nuclear export of TEA domain transcription factor 4 (TEAD4), which otherwise epigenetically modulates nuclear encoded OXPHOS genes to maintain mitochondrial activities [260].

### 4.1. Post-Translational Modifications

Metabolites can also be integrated into gene regulation as substrates for DNA methylation and as post-translational modulators of transcription factors and chromatin modifiers [261,262,263]. Modification status and resulting outputs are orchestrated via coordinated actions on modification marks by writers, erasers, and readers. In many cases, the Michaelis–Menten constants (Km) for these enzymes are higher than metabolic substrates, making them susceptible to changes in metabolite levels [264]. In addition to phosphorylation, acetylation, and other acyl modifications, methylation and O-linked N-acetylglucosamine modification (O-GlcNAcylation) are important post-translational modifications (PTMs) requiring metabolites as substrates [265].

Uridine diphosphate N-acetyl glucosamine (UDP-GlcNAc), which is the end product of the hexosamine biosynthetic pathway (HBP), serves as the substrate for O-linked β-N-acetylglucosamine transferase (O-GlcNAc transferase, OGT). It transfers N-acetylglucosamine to the serine and threonine residues of numerous proteins, including transcription factors and chromatin modifier [266]. High OGT activity is essential for the proliferation of MYC-driven PCa cells, while CRPC reportedly have reduced HBP metabolites, suggesting an intricate metabolic re-wiring process during PCa progression [267,268].

Protein lysine acetylation represents a highly dynamic and reversibly regulated PTM of transcription factors, histones, and chromatin modifiers [269]. Lysine acetyltransferases use acetyl-CoA as a substrate to transfer acetyl groups to the lysine residues of target proteins. Cytoplasmic acetyl-CoA enters the nucleus to serve as the substrate for lysine acetyltransferase. Alternatively, pyruvate, acetate, and citrate diffuse into the nucleus to be converted to acetyl-CoA by the nuclear counterparts of corresponding enzymes [270] (Figure 4A). Nuclear PDH complex (PDC) coordinates lipogenic programs by epigenetically controlling the expression of SREBP target genes through localized acetyl-CoA supply and histone acetylation [156]. On the other hand, protein deacetylases are grouped into zinc-dependent class I, II and IV enzymes, and class III enzymes which require NAD^+^ as a cofactor [271]. The availability of nuclear pools of acetyl-CoA and NAD^+^ partly determines the activities of the corresponding enzymes, thereby linking the general energetic status of the cell to epigenetic and non-epigenetic controls of gene regulation [263]. The Class III enzyme sirtuin (SIRT1) couples epigenetics with cellular redox by utilizing NAD^+^ as a rate-limiting substrate. The nuclear pool of NAD^+^ is regenerated either via the NAD^+^ salvage pathway with the precursor nicotinamide (NAM) in the nucleus, or through nuclear entry of cytoplasmic NAD^+^ [272]. Interestingly, DNA damage leads to PARP1 activation to drastically consume nuclear NAD^+^ for polyADP-ribosylation, presumably outcompeting SIRTs [273]. Aging is associated with declining NAD^+^ levels, reducing SIRT activity [272]. Nevertheless, SIRT1 is overexpressed in multiple cancers, and ADT promotes its nuclear accumulation in PCa models [274]. Nuclear SIRT1 levels correlate with adverse outcomes in PCa. Pharmacologic inhibition of SIRT1 by EX-527 is effective in suppressing the growth of multiple PCa models, suggesting a potential therapeutic strategy in combining SIRT1 inhibition with anti-AR therapy [274].

#### 4.1.1. Lactylation

Recent advancements in analytical tools have led to discovery of various types of lysine acylation by metabolites, including succinylation, malonylation, glutarylation, lactylation, and serotoninylation [275]. Lysine succinylation, malonylation, and glutarylation are physiologically linked and known to be deacylated by Sirt5 [164,276]. However, specific enzymes for many of these modifications are not precisely known, and some reactions may proceed in non-enzymatic manners [275]. Nevertheless, the biological significance of protein lactylation has just begun to be unraveled [277] (Figure 4B). Histone lactylation and acetylation compete for the same modification sites, thus reflecting the cellular metabolic status of whether pyruvate is committed to lactate or acetyl-CoA generation [278]. Glycolytic inhibition and activation decrease and increase lysine lactylation, respectively, while genetic inhibition of LDH abolishes lysine lactylation [279]. This PTM was first discovered in the context of macrophages as part of mechanistic studies investigating how an acidic microenvironment contributes to immunosuppression. It was shown that histone lactylation stimulated gene transcription from the chromatin and induced a homeostatic gene expression program that reverted M1-polarized macrophages back towards a more suppressive, wound-healing, M2-like phenotype characterized by arginase 1 (Arg1) expression [279]. Since then, the immunosuppressive effects of protein lactylation has been demonstrated in additional anticancer immune cell types [280]. For example, lactylation of MOESIN in regulatory T cells (Tregs) enhanced its interaction with TGF-beta receptors TGFBR1 and helped increase FOXP3 expression to mediate its immunosuppressive effects [281]. Furthermore, histone lactylation in tumor-infiltrating myeloid cells also enhanced their immunosuppressive effects via a METTL3-mediated RNA modification [282].

Beyond mediating immunosuppression, the mechanisms and functional effects of histone/protein lactylation are becoming progressively unveiled [283]. For example, it is now known that the acetyltransferase p300 mediates histone lactylation, while HDAC1-3 serves as delactylases [284]. Furthermore, it has been shown that lactylation on the glycolytic enzyme ALDOA serves to allosterically inhibit its enzymatic activity, potentially mediating a negative feedback loop for the glycolysis pathway in the presence of excess lactic acid [283]. Finally, the functional relevance of lactylation has also been reported in RNA splicing, neurodevelopment, osteoblast differentiation, and cancer progression [285,286,287,288]. While there are no published studies of lactylation in PCa at this time, it is likely that this novel PTM will be validated as functionally relevant in promoting PCa progression, especially with respect to having regulatory effects on metabolic pathways and immunosuppressive effects on various components of anticancer immunity.

#### 4.1.2. Methylation

SAM is linked to one-carbon metabolism and is a universal methyl donor for protein, DNA, and RNA modifications [289]. Histone and non-histone proteins undergo protein methylation at arginine and lysine residues. Except for disruptor of telomeric silencing 1 like (DOT1L), lysine methyltransferases (KMT) contain a highly conserved Su(var)3-9, E(z) and Trithorax (SET) domain, examples being the EZH2 and mixed-lineage leukemia (MLL) methyltransferases. Arginine methylation involves the protein arginine methyltransferase (PRMT) family [290]. Demethylases are divided into two subgroups: the flavin adenine dinucleotide (FAD)-dependent lysine-specific demethylases (LSD1 and LSD2) and the Jumonji C domain-containing (JMJD) protein family, which acts on both methylated arginine and lysine residues [291]. Nearly all have been found to be overexpressed in PCa [292,293]. JMJD2 family proteins interact with ETV1 and AR to promote PCa growth [294]. JMJD1a/KDM3A activates AR signaling by upregulating c-MYC expression through H3K9 demethylation and promoting alternative splicing to generate AR-V7 [295,296].

DNA methylation occurs as 5-methylcytosine (m5C) and is controlled by the opposite actions of DNA methyltransferases (DNMTs) and ten-eleven translocation (TET) proteins, which primarily play a demethylation role [297,298]. DNA methylation status is not static during PCa progression [299]. The early and recurrent acquisition of hypermethylation in the glutathione *S*-transferase P1 (GSTP1) promoter has been observed in prostate tumorigenesis [300]. In later stages of the disease, overexpression of the oncogenic drivers AR, MYC, FOXA1, HOXB13, and ERG is associated with hypermethylation at the intergenic regions of these genes and colocalization of H3K27 acetylation [301].

The m6A modification is a reversible PTM of RNA that impacts RNA metabolism, including transcription, pre-mRNA splicing, mRNA export, mRNA stability, and translation [302]. This methylation status is dynamically and reversibly regulated by methyltransferase as writers (METTL3/14, WTAP, RBM15/15B and KIAA1429), demethylases as erasers (FTO/ALKBH5 and ALKBH5), and the binding proteins (YTHDF1/2/3, IGF2BP1 and HNRNPA2B1) as readers. Several reports suggest that METTL3 expression is elevated in PCa, possibly leading to increased m6A RNA methylation. The reader YTHDF2 has also been found to be upregulated in PCa, its expression predicting worse overall survival [303,304,305].

Finally, three key oncometabolites (2-hydroxyglutarate, succinate, and fumarate) drive malignant transformation primarily through methylation regulation and pseudohypoxic signaling. They serve as structural mimetics to α-KG and thus competitively inhibit protein, DNA, and RNA demethylases and proline hydroxylases belonging to the superfamily of α-KG-dependent dioxygenases [44,45]. Indeed, genome-wide methylation landscape studies identified a new epigenetic CpG methylator phenotype (CMP) subtype of mCRPC, which is enriched for mutually exclusive mutations in TET2 and IDH1 [301]. 2-HG producing-mutations in IDH1 are associated with CpG island hyper-methylations in the Cancer Genome Atlas (TCGA) primary PCa database [306]. It remains to be answered how germline SDH and FH mutations exhibit a high degree of tissue specificity in the patients without drastic adverse effects in most other tissues [44,45,307].

## 5. Tumor Microenvironment (TME)

Oxygen and nutrients are supplied from the tumor vasculature, which distributes unevenly in the TME. Both competitive and cooperative metabolic interactions further shape and orchestrate the TME to support metastatic heterogeneity and therapeutic resistance [308,309,310,311]. The TME forms pro-tumorigenic niches, consisting of different cell types such as cancer-associated fibroblasts (CAF), immune cells, and endothelial cells [308,311,312,313,314]. The differential metabolic demands among TME cells result in metabolic competition and symbiosis to support tumor-specific metabolic reprogramming. This causes the release of immunosuppressive metabolites, facilitates tumor progression, and mediates mechanisms of therapeutic resistance [312,314,315,316,317,318]. Exosomes from prostate CAFs are capable of shifting metabolism in recipient tumor cells by suppressing OXPHOS, thereby increasing glycolysis and enhancing glutamine metabolism as a source of intermediate metabolites [319]. Therefore, pharmacological manipulation of TME metabolism has been recognized and ought to be integrated into overall therapeutic strategies [320,321].

### 5.1. Metabolite-Driven Immunosuppression

PCa is occasionally considered to represent the types of cancer that mainly display immune-desert or immune-excluded phenotypes, although still not conclusive [322]. This may be due to a general lack of PD-L1 expression and the presence of immunosuppressive cytokines and metabolites (adenosine, lactate, kynurenine, etc.) within the TME [323,324,325]. Immunometabolism generally dictates the fate of immune cells and their immunological functions [316,326]. Once activated, natural killer (NK) cells and T cells shift their metabolisms from oxidative metabolism to aerobic glycolysis to proliferate and develop effector functions. On the other hand, Treg cells, which repress antitumor immune responses, are primarily dependent on OXPHOS rather than glycolysis for cell persistence and function. Moreover, pro-inflammatory M1 macrophages are characterized by glycolytic metabolism while tumor-associated macrophages (generally defined as anti-inflammatory M2 macrophages) rely on mitochondrial OXPHOS [327]. Highly glycolytic tumors with expression of HK2 and LDHA thus competitively deprive glucose from the TME to impair antitumor immunity and support tumor progression and metastasis in preclinical models of cancers such as sarcoma [328]. Increased tryptophan catabolism by PCa cells produce kynurenine by the concerted actions of indoleamine-2,3-dioxygenase 1 and 2 (IDO1/2) and tryptophan-2,3-dioxygenase (TDO2) [329]. TDO2 expression is elevated in drug resistant PCa tumor tissues [330]. Kynurenine exhibits tumor promoting activities in cell autonomous and non-autonomous manners. In PCa cells, kynurenine is a ligand for the transcription factor aryl hydrocarbon receptor (AhR), promoting its nuclear translocation and cooperation with NF-κB to transactivate C-MYC expression [330,331]. C-MYC upregulates ATP-binding cassette (ABC) transporters and tryptophan transporters, establishing a tryptophan/TDO2/Kyn/AhR/c-Myc loop to sustain drug resistance [330]. Kynurenine also inhibits the activity of NK cells, dendritic cells (DCs), and proliferating T cells, thus providing an immunosuppressive TME [325,329]. Likewise, tryptophan depletion inhibits proliferation and activation of effector T-cells [329]. Conversely, increased IDO expression promotes M2 polarization in tumor associated macrophages [332], and IDO inhibition is a therapeutic strategy that aims to suppress cancer cell proliferation and reactivate antitumor immunity [333]. A phase 2 clinical trial with the IDO inhibitor indoximod (NLG-8189, 1-Methyl-D-tryptophan) (NCT01560923) provided significant evidence of the efficacy of indoximod as an immunometabolic adjuvant in combination with the DC vaccine sipuleucel-T (Provenge^®^) in advanced PCa [334].

Two conditionally essential amino acids, asparagine and glutamine, work in concert to drive tumor growth and metastasis [335,336,337]. When extracellular glutamine levels decline in preclinical models of breast cancer, asparagine becomes essential for supporting de novo glutamine biosynthesis via upregulation of glutamine synthetase (GLUL) [338]. On the other hand, in PCa models, glutamine deprivation in the TME enhances asparagine biosynthesis in CAFs. This occurs via ATF4 activation due to downregulation of the autophagy adaptor protein p62 [339]. CAFs release asparagine to support tumor growth in the glutamine-limited environment.

#### Cancer-Generated Lactic Acid

The extracellular abundance of lactate in the TME is a result of increased intracellular production and MCT4-mediated efflux of lactate. An acidic TME caused by increased lactic acid production can have multi-fold effects on tumorigenesis and progression. These cancer-promoting consequences include enhancing local tissue invasion and distant metastasis, resisting hypoxia and anticancer therapeutics, and stimulating angiogenesis [168,340,341]. Given the multi-faceted role of lactic acid, one aspect that deserves greater attention is its ability to suppress the anticancer immune response [169] (Figure 4B). The recent reappreciation of lactic acid’s functional role in the TME has coincided with the clinical successes of various anticancer immunotherapeutics. As such, there has been an explosion of interest in trying to understand how lactic acid suppresses the anticancer immune response. By this point, almost every conceivable subtype of immune cell is known to be impacted by cancer-generated lactic acid, whether positively or negatively [342,343]. The existing literature on this topic is extensive [169,315,342,344,345], and we will limit the scope of our discussion here to only the major immune contributors. While some of the immunosuppressive mechanisms have already been highlighted in the previous sections (e.g., lactylation and epigenetic changes), other functional impacts are worth considering. In PCa, lactate flux is indeed critical and MCT4 is a potential therapeutic target. Thus, it is reasonable to extrapolate experimental data in non-PCa models to PCa therapeutics.

Discussions of anticancer immunity begin with the cytotoxic effector subtypes, and more particularly, the balance between the effector CD8^+^ cytotoxic T cells and Tregs. Multiple studies have demonstrated how an acidic TME hampers typical cytotoxic T cell function while allowing Tregs to remain relatively unaffected, thus tipping the balance in favor of an immunosuppressive response [346]. For example, given the recent successes of checkpoint inhibitors in anticancer immunotherapy, it is important to note that lactic acid increases PD-1 expression in tumor-associated Tregs through nuclear factor of activated T-cells (NFAT) signaling. The blockade of PD-1 in Tregs further enhances their immunosuppressive functions, leading to tumor escape [347]. Similarly, increased lactate in the TME reduces the efficacy of f vascular endothelial growth factor (VEGF) inhibition by inducing PD-L1 expression in neutrophils, skewing their differentiation towards a tumor promoting N2 phenotype [348]. Alternatively, as mentioned above, it has been shown that Tregs possess a distinct metabolic profile compared to cytotoxic T cells. Treg activity is suppressed under high-glucose conditions but is enhanced in a high-lactic acid environment, partially through an increased availability of biomolecular intermediates necessary for further Treg proliferation [349,350]. Similarly, metabolic competition with tumor cells for available glucose can also negatively impact effector T cells and their ability to eliminate cancer cells [328,351,352]. An acidic environment can also negatively impact the secretion of pro-inflammatory cytokines such as interferon-γ (IFNγ), further reducing the immune system’s ability to mount an effective anticancer response [353,354]. Finally, lactic acid can also alter T cell differentiation away from the cytotoxic phenotypes and induce anergy in the existing cytotoxic T cell population [355,356].

Beyond cytotoxic effector cells, an acidic TME can also negatively affect the anticancer activities of DCs and macrophages in a broad-ranging manner, impacting maturation, antigen presentation, migration, and the secretion of cytokines [357]. For example, lactate itself can be a negative signaling molecule for antigen presenting cells, mediating an overall immune suppression through both GPR81 and GPR132 [358,359,360,361,362]. Lactate can also inhibit DC maturation and antigen presentation, thus preventing an adequate anticancer immune response [363,364]. Increased lactic acid also skews macrophage polarization towards the M2 phenotype partly through epigenetic modulation involving histone lactylation, causing their loss of immunostimulatory functions [362,365,366,367,368]. This is especially mediated through the reduced production of various pro-inflammatory cytokines necessary to elicit an anticancer immune response [369,370,371]. Thus, tumor-specific glycolytic inhibition not only impairs tumor growth and proliferation but also augments antitumor immunity in the TME [328].

### 5.2. Lactate Shuttle

From a purely metabolic perspective, the combination of MCT1 and MCT4 function can facilitate a “symbiotic” interaction that enhances cancer cell survival. Known as the “lactate shuttle”, it has been suggested that adjacent cancer (and stromal) cells can coordinate their nutrient uptake and processing in order to achieve greater metabolic efficiency [372] (Figure 4C). In a nutrient/oxygen-starved environment, especially towards the interior of the tumor, glucose is metabolized via glycolysis to produce lactic acid. This lactic acid is excreted primarily through the lactate exporter MCT4 and is taken up via MCT1 by more oxidative cancer cells, especially those in a nutrient/oxygen-rich environment. Lactate is converted back into pyruvate in these oxidative cells through a “reversed Warburg effect”, and energy metabolism proceeds via the typical TCA cycle and OXPHOS [373]. There is some indication that this metabolic coupling, specifically between MCT1-expressing cancer cells and MCT4-expressing stromal cells, can help fuel PCa growth [374,375,376]. This tumor metabolic symbiosis requires the specific settings where CAF is glycolytic and tumor cells continuously use lactate as a fuel under complete aerobic conditions [377].

### 5.3. Fatty Acid Signaling

Adipocytes are known to play a role in carcinogenesis, and epidemiological studies have shown a correlation between obesity and cancer incidence, as well as worse prognosis in patients with excess weight [378]. Particularly for PCa, adipocytes from the periprostatic adipose tissue (PPAT) surrounding the prostate gland can support prostate epithelial cancer cell migration through chemokine modulation [379]. PCa cells stimulate these adipocytes to undergo metabolic rewiring, promoting lipolysis and the release metabolic products such as free FAs (FFAs) that serve as the source of FAO in tumor cells. Lipid transfer proteins FABP4 and/or CD36 have been reported to mediate the transfer of FAs from adipocytes into cancer cells including PCa [380]. Pharmacological inhibition of adipocyte lipolysis or lipid transfer may provide effective antitumor activities.

Excess dietary carbohydrates can be converted into lipid through DNL and accumulate into lipid droplets. Dietary lipids can be stored in lipid droplets as well. Regardless of the origin, nearly all cell types can store fat in lipid droplets; adipocytes, however, are specialized cells known for the controlled mechanism of FA accumulation and utilization through lipolysis [381]. Increased FASN expression has been reported in adipose tissue and inversely correlates with insulin sensitivity [382]. As a reservoir of FA, adipocytes are an important energetic support for highly proliferative tumor cells. AR antagonism induces reprograming of lipid metabolism through lowered FASN expression, thereby decreasing intracellular saturated FA levels and promoting uptake of nutrient derived polyunsaturated FAs (PUFAs) in PCa [245,383,384]. The rise in PUFA enhances membrane fluidity and lipid peroxidation, causing hypersensitivity to glutathione peroxidase 4 (GPX4) inhibition and ferroptosis in PCa [384]. Dietary FA also shapes immunity in TME. Uptake of FAs support Treg and memory T cells to drive FAO and OXPHOS for their survival and function [314,385]. Moreover, metabolic fitness and plasticity allow CD8^+^ T cells to upregulate FA catabolism to support their effector functions when glucose is not available [314].

Another key role of lipids in the TME is the pro-inflammatory and pro-tumorigenic function of oxidized FAs such as eicosanoid species in various cancer models, including PCa. There are several types of eicosanoids, including prostaglandins, thromboxanes, leukotrienes, lipoxins, resolvins, and eoxins, and they are all derived from PUFA. Prostaglandins can be produced from arachidonic acid metabolism through the enzymatic activity of cyclooxygenase (COX). Prostaglandins have been implicated in carcinogenesis, and studies show that COX inhibition by non-steroidal anti-inflammatory drugs (NSAIDs) can help delay or prevent PCa [386,387]. Interestingly, prostaglandins have also been shown to impair NK cells viability and activity, modulating the antitumor immune response [388]. Leukotrienes can be obtained by the action of lipoxygenases (LOX) on arachidonic acid. These molecules have also been correlated with carcinogenesis, and overexpression of LOX5 was observed to be upregulated in prostate adenocarcinoma in comparison to benign tissue [389].

## 6. Therapeutic Interventions

A number of factors must be considered when developing metabolism-based therapeutic interventions. To start, the therapeutic targeting of metabolic complexes or enzymes in many cases affects not only tumor cells but also normal proliferating cells [35,46,320]. Multiple enzymatic isoforms can also perform overlapping functions, and metabolic flexibility can divert processes away from inhibited pathways. Therefore, redundancy among metabolic genes could shape the landscape of positive and negative impact on targeting tumor metabolism. A better understanding of the different alterations that facilitate a cancer’s unique metabolic landscape can help ensure efficacy and avoid generic toxicity. Beside the careful selection of therapeutic targets, the fact that metabolites generally require specialized membrane transporters for influx and efflux presents another unique therapeutic challenge. Furthermore, common nutrients such as glucose, glutamine, or FAs can also be utilized through metabolic symbiosis and competition occurring between the tumor cells and the stroma cells in the TME [311,312,313,318]. Thus, interference with these intercellular metabolic interactions can elicit antitumor effects and overcome therapeutic resistance. When these factors are considered collectively, in order to minimize potential toxicity and increase specificity, it is necessary to identify selective metabolic vulnerabilities within a particular cancer in a manner that is associated with specific oncogenic activations. Currently, there are only a limited number of such drugs which are immediately applicable to PCa treatment [51,55]. However, we can take advantage of the potentially effective metabolism-targeted therapeutics being developed in the preclinical setting, as well as those with some clinical trial data for various diseases. This can include indications for non-prostate cancers, diabetes, and infectious diseases as shown in Table 1 (Indications for PCa are shaded) [46,320]. As stated above, various types of cancers share alterations in some metabolic pathways such as fatty acid synthesis, glutamine addiction, and mitochondrial metabolism. One can speculate that corresponding therapeutic agents in clinical trials (Table 1B) will be readily applicable for PCa. Indeed, drug repurposing is becoming one of the most promising strategies for PCa therapeutics due to the high safety profile of established drugs such as metformin and statins [390,391].

There are two classes of drug development platforms which begin with either target-based or phenotypic-based screening of compound libraries [392,393,394]. Typical molecular target-based drug screening programs involve identification of specific targets for a disease of interest, and is followed by in silico or experimental screening of compounds and subsequent chemical optimization of lead candidates for further preclinical testing. Transporters are a notoriously difficult class of therapeutic targets due to the presence of multiple transmembrane domains needed to form the central channel. Such channels are often difficult to undergo crystallization for X-ray structural determination, thus limiting the amount of information available for drug design. However, recent advances in drug development approaches and structural biology have offered unique opportunities to overcome such hurdles. For example, the increasing maturity of cryo-electron microscopy (cryo-EM) techniques have allowed the structures of various important membrane transporters to be determined, including LAT1, LAT2, MCT1, MCT2, and GLUT4 [395,396,397,398,399]. These structures provide critical information for accurately designing and studying potential inhibitors. Alternatively, in our experience with designing selective MCT4 inhibitors, using a computer-assisted drug discovery platform can also be an effective approach [400,401]. Recent developments in big data, graphics processing unit (GPU)-computing, and deep learning has advanced drug discovery into a new artificial intelligence era capable of accurately calculating drug-like properties and binding [400,402,403,404]. Multiple state-of-the-art drug design software and molecular docking programs are now available [405,406,407], and these can be used in combination with advanced structural prediction algorithms such as AlphaFold to arrive at potential therapeutic compounds, especially for difficult drug targets that lack available structural information [408]. Another example includes the integrated platform of high-throughput virtual screening and graphite dots–assisted laser desorption/ionization mass spectrometry for drug discovery (GLMSD), allowing for rapid screening and identification of five potent HK2 inhibitors from initial 240,000 compounds [409].

In contrast, phenotypic screening is aimed at perturbing the biological process without prior understanding of the molecular mechanism of action. This approach requires subsequent steps involving target deconvolution, which can be a challenging and lengthy endeavor. Indeed, direct interaction with a single target is not necessarily responsible for phenotypes observed in screenings. Rather, observed phenotypes may reflect the superposition of polypharmacological effects. Nevertheless, the benefit of an unbiased approach is underscored by the discovery of new therapeutic targets such as FKBP12, mTOR, and histone deacetylases [394]. There is a growing interest in reinventing phenotypic screens with the advent of new tools such as high content imaging, RNA profiling, and CRISPR/Cas9-based genome engineering as discovery platforms [410]. For example, metabolism-focused CRISPR genetic screens revealed tumor-specific metabolic vulnerabilities, thus identifying potential druggable targets [411]. Recently established yeast with full humanization of the glycolytic pathway can also be used as both a target- and a phenotypic-based drug discovery platform, thus integrating traditional high throughput screening for identifying effective anti-glycolytic agents [412].

Isoform specific contribution to tumor progression is exemplified by HK2. Its deficiency is therapeutic despite continued HK1 expression, thus underscoring its potential as a therapeutic target [413,414]. HK2 is required for survival of PTEN/p53-deficient CRPC tumors [127]. Aerobic glycolysis not only supports glycolytic tumors but also drives effector T cell differentiation and proliferation [326,328]. Whereas increased HK2 expression is associated with the Warburg effect in tumors, HK2 is dispensable for T-cell dependent immunity [126,127,415,416]. Moreover, silencing or deletion of specific isozymes results in a therapeutic liability that can be exploited in preclinical cancer models. With silenced HK1, HK2 positive tumor cells become more sensitive to HK2 inhibition [417,418]. As another example of isoform specificity, TEPP-46 binds to PKM2 but not PKM1 at an allosteric site distinct from the 1,6-FBP site, which is required for the formation of a tight tetrameric enzyme [419,420]. PKM2 is abundantly found as a dimer in proliferating cells. The lower enzymatic activity of PKM2 compared to PKM1 allows cancer cells to rewire their metabolism and utilize the intermediates of glycolysis for biosynthetic pathways [133]. Consistently, pharmacological activation of PKM2 by TEPP-46 suppresses tumor progression in PCa models [134]. Finally, passenger deletion of enolase 1 (ENO1) is associated with homozygous deletion of the 1p36 tumor-suppressor locus, which provides selective vulnerability to a potent prodrug of the competitive ENO2 inhibitor, POMHEX [421,422].

Targeting glutamine addiction is attractive in cancer therapeutics [423,424]. Clinically tested CB-839 is a selective allosteric inhibitor for GLS1 without discrimination between KGA and GAC and spares GLS2 [425]. GLS1 inhibition may offer clinical benefits in advanced and therapy-resistant PCa overexpressing the GAC isoform, which is sensitive to GLS1 inhibition by CB-839 [180]. CB-839 has been in clinical trials for various cancer types as a monotherapy and in combination with an immune checkpoint inhibitor (Table 1). Furthermore, pan-inhibition of glutamine metabolism in the TME provides therapeutical benefits in the context of cancer immunotherapy. Pro-drug JHU-083 is activated in the TME to release the glutamine analog 6-diazo-5-oxo-L-norleucine (DON), which elicits covalent inhibition of multiple glutamine-utilizing enzymes [426]. Warburg-type cancer cells direct pyruvate away from mitochondria and rely on glutamine as an anaplerotic substrate to replenish TCA cycle intermediates. Glucose deficiency or excessive lactate in TME can create glutamine dependency in effector T cells and NK cells for their immune functions. Therefore, JHU-083 administration presumably leads to suppression of tumor cells and blunts the immune competent environment. JHU-083 simultaneously inhibits glycolysis and OXPHOS, thereby comprehensively disintegrating bioenergetics in tumor cells. Nevertheless, JHU-083 augments oxidative phenotypes in effector CD8^+^ T cells, which upregulates ACSS2 to generate high levels of acetyl-CoA to fuel the TCA cycle. Glutamine-based metabolic symbiosis is also observed in PCa. Thus, pan-inhibition of glutamine metabolism may also offer promising therapeutic benefits in PCa.

As a hallmark of cancer, lipid metabolism has been investigated as a therapeutic target to treat several types of tumors. Particularly, FASN inhibitors have been developed and characterized in several preclinical models, including in PCa. First-generation FASN inhibitors, such as C75, orlistat, and cerulenin, successfully reduced cancer cell growth in preclinical models [427,428,429]. However, an undesirable side-effect profile and a poor pharmacological response have prevented such molecules from clinical usage in cancer therapy. C75 and cerulenin caused reduced food intake and a profound weight and adipose mass loss [430]. C75 was also observed to increase FAO through CPT1 activity stimulation, further impairing energetic metabolism [431]. Despite therapeutic implications for obesity, pharmacologically induced weight loss is threatening for cancer patients that commonly experience cachexia. Newer molecules targeting FASN with promising results in preclinical models have been recently developed, such as GSK2194069 [432], Fasnall [433], IPI-9119 [86] and BI99179 [434]. However, so far, only one FASN inhibitor has entered clinical trials (TVB-2640 from Sagimet Biosciences) as shown in Table 1B. This compound has demonstrated potent FASN inhibition as well as a favorable and manageable safety profile, characterized by very limited, non-serious, reversable adverse events when tested in patients with previously treated advanced metastatic solid tumors, including 4 PCa patients [435]. Particularly in PCa, FASN inhibition leads to reduced cell growth, increased apoptosis, cell cycle arrest and down-regulation of both full-length AR and splice variants in vitro and in vivo, as well as suppression of AR-V7-driven gene-sets and modulation of both AR and AR-V7 nuclear translocation [86]. Because of its effects on AR/AR-V7 as known drivers of ADT resistance, inhibitors of FASN can be used to sensitize cells to AR antagonists to delay or overcome therapy resistance. Thus, targeting lipid synthesis represents a novel therapeutic strategy to target AR-dependent CRPC.

Mutant-specific inhibitors provide effective anticancer therapies through preferential inhibition of oncogenic signaling without compromising normal cell functions. A mutant specific BRAF inhibitor, vemurafenib, specifically targets oncogenic MAPK signaling associated with BRAF V600mut for effective targeted therapies [436]. Imatinib and gefitinib improve the outcomes of patients with tumors harboring BCR-ABL fusion and EGFR mutation, respectively, [437]. Besides SDH and FH mutations, isocitrate dehydrogenase 1 (IDH1) and 2 (IDH2) mutations are the only known oncogenic mutations of metabolic enzymes [438]. 2-HG is virtually absent in normal tissues, unlike the two other well-characterized oncometabolites succinate and fumarate. 2-HG has been reported to reach millimolar concentrations in tumors with mutant IDH1 and IDH2 [439]. Ivosidenib (AG-120) and enasidenib (AG-221) allosterically inhibit these neomorphic functions of mutant IDH1 and IDH2, respectively, [438]. Targeting mutant IDH is attractive but limited in PCa: IDH1 mutations account for only 0.3–2.7% of PCa incidence, while neomorphic IDH2 mutations are not reported [440].

The recent clinical success of PARP inhibitors in BRCA-less cancers led researchers to explore other synthetic lethal interactions [441,442]. Development of genome-wide RNAi-mediated knockdown and CRISPR knockout screens allowed researchers to discover cancer-specific metabolic vulnerabilities, including synthetic lethal interactions between *PRMT5* inhibition and *MTAP* deletion [443,444]. Furthermore, simultaneous perturbation of functional paralogs led to the identification of the dual deletion of dual-specificity phosphatase DUSP4 and DUSP6 as digenic synthetic lethal targets in BRAF- and NRAS-mutant melanoma with persistent hyperactivation of MAPK signaling [445]. These approaches taken in non-PCa cases can be simply extended to PCa studies. Effective new targets relevant to the most common oncogenic drivers (PTEN, P53, MYC) remain to be uncovered [442].

Increased demand for exogenous supply of amino acids causes auxotrophy for non-essential amino acids [446]. Application of therapeutic enzymes is an emerging field of research and leads to systemic depletion of specific metabolites, especially those that tumors cells are addicted to. Asparaginase was the first Food and Drug Administration (FDA)-approved amino acid degrading enzyme used to treat acute lymphoblastic leukemia (ALL) [447]. Depletion of arginine by administration of recombinant proteins such as the human arginase and the bacterial arginine deiminase is being explored for the treatment of arginine-auxotrophic tumors, including ASS1-deficient PCa [448]. Introduction of cysteinase suppressed tumor cell proliferation in patient-derived xenografts (PDXs) derived from PCa [449]. Dietary restriction of amino acids represents another strategy for selective auxotrophic tumors [450]. Serine starvation is effective in p53-deficient tumors, while glycine deprivation is well fitted to treat rapid cancer cell proliferation [204,451,452].

**Table 1 biomolecules-12-01590-t001:** Agents targeting metabolism in preclinical and clinical studies. (**A**) Preclinical studies; (**B**) Clinical studies.

(**A**)					
**Target**	**Agent**	**Modes of Action**		**Study Design and Results**	**Reference**
Glycolysis					
PKM2	TEPP-46	Small molecule pyruvate kinase activators have been identified that stabilize the PKM2 tetramer to promote an enzyme state similar to PKM1.		Activation of PKM2 led to tumor inhibition in mice with prostate specific deletion of PTEN.	[420]
LDHA	GSK2837808A	A potent, selective and NADH-competitive inhibitor of LDH-A.		Combination therapy with radiation improved antitumoral T-cell response and reduced tumor progression in pancreatic cancer models.	[453,454]
MCT1	AR-C122982/SR13800	Selective MCT1 inhibitor.		Antitumor effects on MYC-overexpressing breast cancer and neuroblastoma in preclinical models.	[455,456]
Mitochondrial metabolism					
MPC	MSDC-0160	Thiazolidinedione that reversibly binds to MPC1 and 2 heterodimer.		Inhibition of tumor growth in preclinical models of both castration-sensitive and resistant AR-driven prostate tumors.	[147,457]
MPC	UK-5099	Inhibits the MPC complex by binding to Cys54 of MPC2 in a covalent manner.		Inhibition of tumor growth in preclinical models of both castration-sensitive and resistant AR-driven prostate tumors.	[458]
Complex V	Gboxin	Inhibits Complex V (F0F1-ATP synthase).		Antitumorigenic effects on in vivo and in vitro glioblastoma model.	[459]
Amino acid metabolism					
ASCT2	V-9302	Competitive small molecule antagonist of ASCT2, leading to decrease in glutamine influx.		Antitumor effects on preclinical models including xenografts from triple negative breast cancer cell line HCC-1806.	[460]
				Improves antitumor T cell activity in triple-negative breast cancer.	[461]
GLS1 (KGA and GAC)	CB-839/Telaglenastat	Orally available potent allosteric inhibitor for GLS1.		Antitumor effects on triple negative breast cancer models and lung tumor models.	[425,462,463]
IDO1	BMS-986205	Inhibits apo-IDO1 binding to heme which is co-factor for IDO1 catalytic action.		Iinrodostat potently and specifically inhibits IDO1 to block an immunosuppressive mechanism that could be responsible for tumor escape from host immune surveillance with favorable PK/PD characteristics that support clinical development.	[464]
Fatty acid metabolism					
SREBP	Fatostatin	Prevent nuclear translocation of SREBPs by inhibiting the SREBP cleavage-activating protein (SCAP) that facilitates the ER-Golgi translocation of SREBPs.		Inhibition of SREBP and AR transcription networks is associated with antitumorigenic effects on in vitro and in vivo PCa models.	[465,466]
FASN	IPI-9119	Inhibits the FASN thioesterase domain by promoting acylation of the catalytic serine.		Tumor inhibition is associated with altered fatty acid metabolism and reduced AR signaling.	[86]
ACC1 and ACC2	ND-654	Allosteric inhibitor of the ACC (Acetyl-CoA carboxylase) enzymes that prevents dimerization of ACC.		Inhibition of de novo lipogenesis in liver and the development of hepatocellular carcinoma.	[467]
ACC1 and ACC2	ND-646	Allosteric inhibitor of the ACC (Acetyl-CoA carboxylase) enzymes that prevents dimerization of ACC.		Suppression of lung tumor growth in the mouse models of non-small cell lung cancer.	[468]
(**B**)					
**Target**	**Agent**	**Modes of Action**	**Clinical Relevance (NCT Number, Study Phase, etc.)**	**Study Design**	**Reference**
Glycolysis					
MCT1 and MCT2	AZD3965	A potent pyrimidine-derived inhibitor of MCT1 with activity against MCT2 but selectivity over MCT3 and MCT4.	NCT01791595 (Phase 1)	Trial of AZD3965 in patients with advanced cancer.	[469,470,471,472]
Mitochondrial metabolism					
PDH/alfa-KGDH	CPI-613/devimistat	Lipoate analog that inhibits two lipoate-dependent enzymes, ketoglutarate dehydrogenase (KGDH) and pyruvate dehydrogenase (PDH).	NCT03504423 (Phase 3)	Study evaluating efficacy and safety of thymidylate synthase inhibitor FFX versus combination of CPI-613 with mFFX (modified FFX) in patients with metastatic adenocarcinoma of the pancreas.	[473,474,475,476,477]
			NCT03435289 (Phase 1)	CPI-613 with nucleoside metabolic inhibitor Gemcitabine and albumin-bound paclitaxel Nab-paclitaxel for patients with advanced or metastatic pancreatic cancer.	
mutant IDH2	Enasidenib/AG-221	Allosteric binding of AG-221 stabilizes inhibitory open conformation of the mutant IDH2.	FDA approval	Treatment of relapsed or refractory acute myeloid leukemia (AML).	[478]
Complex I, PEN2, polypharmacologic	Metformin	Inhibits complex I to decrease OXPHOS activity. Metformin binds to PEN2 (presenilin 2) to form a complex with ATP6AP1, a subunit of the v-ATPase8 and inhibit v-ATPase.	NCT02614859 (Phase 2)	Bicalutamide with or without metformin for biochemical recurrence in overweight or obese PCa patients (BIMET-1).	[479,480,481,482,483]
			NCT02339168 (Phase 1)	Enzalutamide and metformin hydrochloride in treating patients with hormone-resistant PCa.	
			NCT03291938 (Phase 1)	Test in advanced cancers.	
			NCT02339168 (Phase 1)	Enzalutamide and metformin hydrochloride in treating patients with hormone-resistant PCa.	
Complex I	IACS-010759	Binds to inhibit complex I (NADH ubiquinone oxidoreductase).	NCT02882321 (Phase 1)	Treating patients with relapsed or refractory acute myeloid leukemia (AML).	[484,485]
Complex I	IM156	Orally available metformin derivative.	NCT03272256 (Phase 1)	Patients with advanced solid tumor and lymphoma.	[486,487,488]
Amino acid metabolism					
LAT1	JPH203	Selective L-type amino acid transporter 1 inhibitor.	UMIN000016546 (Clinical trial registration in Japan)	Exploratory analyses of biomarker using change of blood free amino-acid concentration related to JPH203 (LAT1 inhibitor).	[489,490,491]
GLS1 (KGA and GAC)	CB-839/Telaglenastat	Orally available potent allosteric inhibitor for GLS1.	NCT03875313 (Phase 1b/2)	Combination with PARP inhibitor Talazoparib in patients with solid tumors.	[425,462,492]
			NCT04250545 (Phase 1)	To determine the safety and tolerability of CB-839 combination with mTORC1/2 inhibitor MLN0128 (sapanisertib) and determine the recommended phase 2 dose (RP2D) of the combination.	
GLS1 (KGA and GAC)	IPN60090	Orally available potent allosteric inhibitor for GLS1.	NCT03894540 (Phase 1)	Dose escalation and dose expansion study of IPN60090 in patients with advanced solid tumors.	[493]
Glutamine metabolism	Sirpiglenastat/DRP-104	Pro-drug form of a broad acting glutamine antagonist that irreversibly inhibits multiple enzymes involved in glutamine metabolism. Different from JHU083.	NCT04471415 (Phase1/2)	As single agent and in combination with checkpoint inhibitor atezolizumab in patients with advanced solid tumors.	[494]
IDO1	Epacadostat/INCB024360	Orally available reversible competitive potent IDO1 inhibitor.	NCT03516708 (Phase 1)	Epacadostat (INCB024360) added to preoperative chemoradiation in patients with locally advanced rectal cancer.	[495]
			NCT03589651 (Phase 1)	Checkpoint inhibitor INCMGA00012 in combination with other therapies in patients with advanced solid tumors.	[494]
IDO1	Indoximod	Orally available reversible competitive potent IDO1 inhibitor.	NCT02073123 (Phase 1/2)	IDO inhibitor in combination with checkpoint inhibitors for adult patients with metastatic melanoma.	[334,464]
			NCT04049669 (Phase 2)	Indoximod with chemotherapy and radiation for relapsed brain tumors or newly diagnosed diffuse intrinsic pontine gliomas (DIPG).	
			NCT01560923 (Phase 2)	Combined therapy with Sipuleucel-T and Indoximod for patients with refractory metastatic PCa.	[334]
MAT2A	AG-270	Allosterically inhibits MAT2A (methionine adenosyltransferase 2A) activity by preventing product release, leading to decrease in intracellular SAM levels.	NCT03435250 (Phase 1)	Participants with advanced solid tumors or lymphoma With MTAP loss.	[496,497]
Fatty acid metabolism					
HMG-CoA reductase	statin, Atorvastatin	Competitive inhibitor for HMG-CoA reductase to decrease cholesterol level.	NCT04026230 (Phase 3)	Atorvastatin on PCa progression during ADT (ESTO2).	[390,391,498]
			NCT02003924 (Phase 3), PROSPER	Safety and efficacy study of enzalutamide in patients with nonmetastatic CRPC.	
			NCT01212991 (Phase 3), PREVAIL	Safety and efficacy study of oral MDV3100 in chemotherapy-naive patients with progressive metastatic PCa.	
			NCT00974311 (Phase 3), AFFIRM	Safety and efficacy study of MDV3100 in patients with CRPC who have been previously treated with docetaxel-based chemotherapy.	
				Retrospective analyses of all three trials pooled and AFFIRM + PREVAIL pooled revealed that statin but not metformin use was significantly associated with a reduced risk of death in enzalutamide-treated patients.	
FASN	TVB-2640	Inhibits the β-ketoacyl-ACP reductase activity of FASN.	NCT03179904 (Phase 2)	TVB-2640 and Trastuzumab (herceptin) in combination with paclitaxel or endocrine therapy for the treatment of HER2 positive metastatic breast cancer.	[499,500]
			NCT03808558 (Phase 2)	TVB-2640 in KRAS non-small cell lung carcinomas.	
ACC1 and ACC2	GS-0976/ND-630/NDI 010976/Firsocostat	Allosteric inhibitor of the ACC (Acetyl-CoA carboxylase) enzymes that prevents dimerization of ACC.	NCT03449446 (Phase 1)	Study to evaluate the safety and efficacy of Selonsertib, Firsocostat, Cilofexor, and combinations in participants with bridging fibrosis or compensated cirrhosis due to nonalcoholic steatohepatitis (NASH).	[501,502]
Another metabolism					
mutant IDH1	Ivosidenib/AG-120	Allosteric binding of Ivosidenib stabilizes inhibitory open conformation of the mutant IDH1 active site.	FDA approval	Newly diagnosed acute myeloid leukemia (AML) with a susceptible IDH1 mutation.	[503]
Sirt1	EX-527/Selisistat/SEN0014196	Stabilize the closed enzyme conformation preventing product release.	NCT04184323 (Phase 2) terminated due to lack to funding	SIRT-1 antagonism for endometrial receptivity.	[504]
				This compound has reached Phase 2 clinical trials for Huntington’s disease (HD) found to be safe and well tolerated in early HD patients at plasma levels within the therapeutic concentration range in preclinical HD models.	[505]
DHODH	Leflunomide	Competitive inhibitor for mitochondrial enzyme DHODH.	NCT03709446 (Phase1/2)	Treating patients with previously treated metastatic triple negative breast cancer.	[506,507]
			NCT04997993 (Phase 1)	Patients with PTEN-null advanced solid malignancies.	
ODC1	2-difluoromethyl ornithine (DFMO)/eflornithine	Competitive and irreversible inhibitor of ODC1.	NCT04301843 (Phase 2)	Eflornithine (DFMO) and etoposide for relapsed/refractory neuroblastoma.	[508,509]
			NCT01349881 (Phase 3)	Adenoma and second primary prevention trial.	

## 7. Concluding Remarks

There is a growing number of promising biomarkers, such as AR, SPOP, PTEN, AKT, RB1, BRCA1/2, PSMA, and CDK12, to inform treatment decisions for PCa. Despite considerable efforts with significant success, precision oncology in advanced PCa had to face challenges, including drug resistance and limited population who benefit from the related therapeutics [48].

Tumor metabolism has emerged as a highly attractive therapeutic target for a single therapy or in combination with chemotherapy and immunotherapy [35,39,41,46]. The recent FDA approval of inhibitors for IDH neomorphic mutants represents a milestone achievement in precision oncology in the context of cancer metabolism [438]. Understanding the multi-faceted metabolic interactions will help discover and develop reliable diagnostic and predictive markers for precision oncology applications [51,52,54,55]. Currently, however, there are a limited number of reliable metabolic biomarkers, including PET-based tumor imaging, that characterizes related metabolic pathways and bioenergetics, alterations in metabolic genes, and circulating oncometabolites.

Advancement in analytical technologies allow us to characterize tumor metabolism in different aspects. Matrix-assisted laser desorption/ionization mass spectrometry imaging (MALDI MSI) allows for spatial differentiation of metabolism in PCa tissue [510]. Though still challenging, single-cell metabolomics can be potentially used to optimize metabolome-based treatment decisions [511]. Tumor cells may rewire metabolic pathways to develop resistance to specific metabolic interference. In this regard, simultaneous blockade of multiple pathways may offer advantageous therapy as exemplified by JHU-083 [426]. CRISPR based genome-wide knockout screens have incredible opportunities to accelerate identification of synthetic lethal pairings for combined therapies targeting metabolism [442].

We have described the metabolic phenotypes, networks, and interactions in relation to the disease stages and the TME during PCa progression. In addition to AR, oncogenic drivers display distinct dominant metabolic dependencies. To develop effective metabolism-targeting precision therapies, it is crucial to further identify the unique metabolic signatures that define and support the malignant phenotypes of PCa, particularly that of CRPC.

## Figures and Tables

**Figure 1 biomolecules-12-01590-f001:**
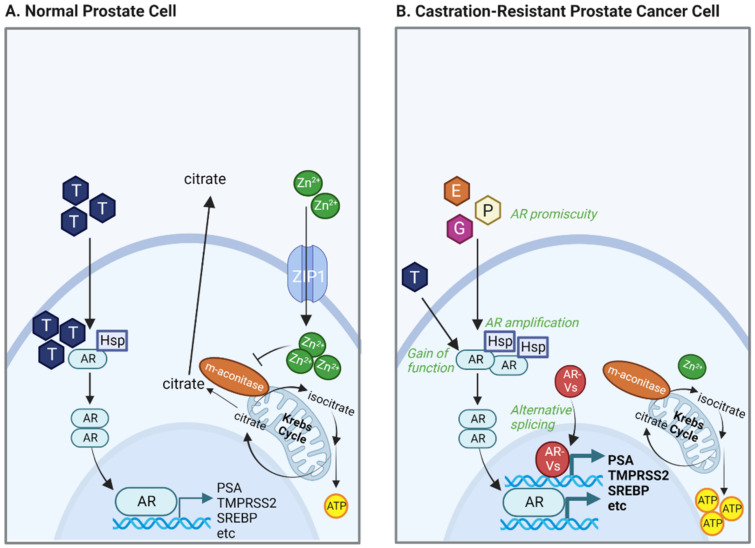
Metabolic reprogramming involved in PCa progression to CRPC. Alterations in Zinc-driven metabolism and AR pathway are observed as prostate cancer progresses to CRPC. Mechanisms of androgen resistance include AR promiscuity, amplification, gain of function and alternative splicing (Created with BioRender.com). T—testosterone; E—estrogen; P—progesterone; G—glucocorticoids.

**Figure 2 biomolecules-12-01590-f002:**
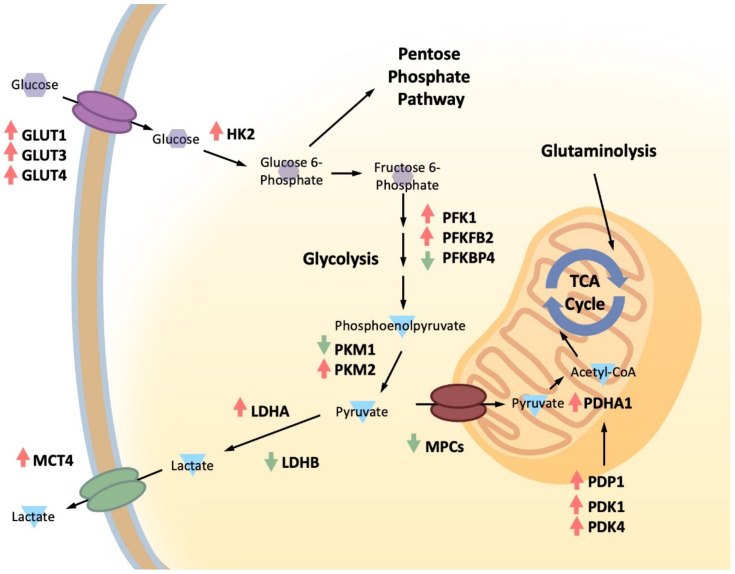
Altered glucose metabolism and the reported aberrations that help facilitate an increased glycolytic phenotype in aggressive PCa cells.

**Figure 3 biomolecules-12-01590-f003:**
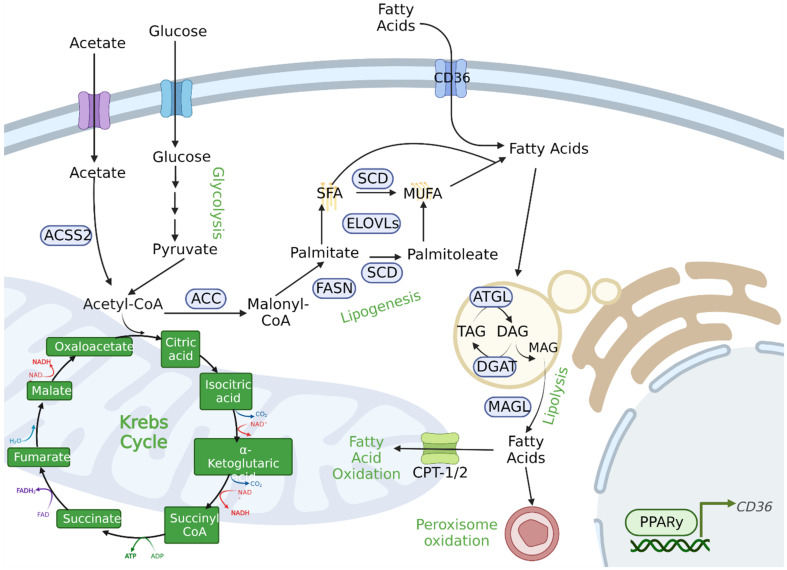
Major lipid metabolism enzymes and metabolites that are modulated on PCa cells. Several alterations are observed during prostate carcinogenesis, including increased rates of de novo lipogenesis, increased fatty acid uptake, lipolysis and fatty acid oxidation (Created with BioRender.com).

**Figure 4 biomolecules-12-01590-f004:**
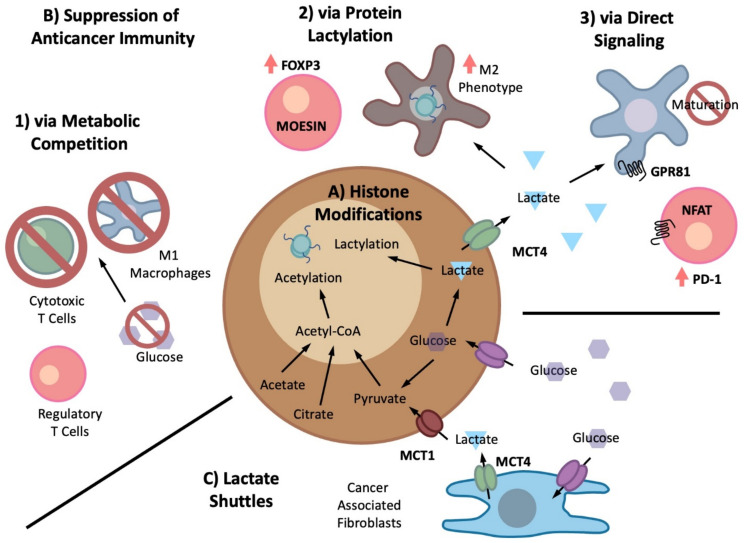
Altered glucose metabolism in aggressive PCa and the effects of increased lactic acid production on (**A**) gene regulation via histone modifications, and (**B**,**C**) the TME, especially its impacts on suppressing the anticancer immune response.

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
