# Peer review of "Druggable Metabolic Vulnerabilities Are Exposed and Masked during Progression to Castration Resistant Prostate Cancer"

_biomolecules, 2022, doi:10.3390/biom12111590_

Round 1

Reviewer 1 Report

I enjoyed reading this comprehensive review of prostate cancer metabolism and potential metabolism-related therapeutic targets and think it is valuable and would be of interest to the readers of this journal. Its breadth (>500 references!) is a strength, as it could function as a broad overview or introduction to the field, an encyclopedia to find references for the reader to learn more depth, which is certainly inherently useful. However some sections might be improved by focusing a bit - there are some parts that seem to discuss tumor metabolism in general instead of sticking to focus on PCa. For example the TME section is not clear what data is derived from PCa tumors and models versus other tumor types. Similarly the table seems to be about agents targeting cancer metabolism in any cancer type - it could be improved by telling the reader what was studied in PCa and what is being seriously considered or is already in clinical trials for PCa.

Minor points:

- ACC is thought to be the rate limiting enzyme for FA synthesis, not FASN - this is mentioned in 2 locations

- Section 5.1 statement about why PCa is immune cell-excluded is too definitive given this is really not known 

Author Response

Thank you so much for taking the time to review our manuscript and share your thoughtful comments. We added changes in red to this revised manuscript. We hope that we adequately addressed your concerns to the best of our ability.

I enjoyed reading this comprehensive review of prostate cancer metabolism and potential metabolism-related therapeutic targets and think it is valuable and would be of interest to the readers of this journal. Its breadth (>500 references!) is a strength, as it could function as a broad overview or introduction to the field, an encyclopedia to find references for the reader to learn more depth, which is certainly inherently useful.

Thank you for these excellent comments.

However some sections might be improved by focusing a bit - there are some parts that seem to discuss tumor metabolism in general instead of sticking to focus on PCa. For example the TME section is not clear what data is derived from PCa tumors and models versus other tumor types.

To clarify whether PCa is the subject or not, we added “types of cancer (PCa, sarcoma, breast cancer)” in the text if necessary. To emphasize our focus is PCa, we also added sentences such as “In PCa, lactate flux is indeed critical and MCT4 is a potential therapeutic target. Thus, it is reasonable to extrapolate experimental data in non-PCa models to PCa therapeutics.”

Similarly the table seems to be about agents targeting cancer metabolism in any cancer type - it could be improved by telling the reader what was studied in PCa and what is being seriously considered or is already in clinical trials for PCa.

To distinguish indications for PCa from non-PCa , we shaded table rows corresponding to PCa cases. To emphasize our focus is PCa, we also added sentences such as “As stated above, various types of cancers share alterations in some metabolic pathways such as fatty acid synthesis, glutamine addiction, and mitochondrial metabolism. One can speculate that corresponding therapeutic agents in clinical trials (Table 1B) will be readily applicable for PCa.”, “Glutamine-based metabolic symbiosis is also observed in PCa. Thus, pan-inhibition of glutamine metabolism may also offer promising therapeutic benefits in PCa.”, and “These approaches taken in non-PCa cases can be simply  extended to  PCa studies.

ACC is thought to be the rate limiting enzyme for FA synthesis, not FASN - this is mentioned in 2 locations

Thank you for correcting us. We elected to use “principal” instead of “rate limiting”.

Section 5.1 statement about why PCa is immune cell-excluded is too definitive given this is really not known 

Thank you again for correcting us. To avoid definitive statements, we changed sentences as follows: “PCa is occasionally considered to represent the types of cancer that mainly display immune-desert or immune-excluded phenotypes, although still not conclusive [326]. This may be due to a general lack of PD-L1 expression and the presence of immunosuppressive cytokines and metabolites (adenosine, lactate, kynurenine, etc.) within the TME [327-329].” We added the reference [326] which discusses unique immune microenvironment of PCa. 

[326] Stultz, J. and L. Fong, How to turn up the heat on the cold immune microenvironment of metastatic prostate cancer. Prostate Cancer Prostatic Dis, 2021. 24(3): p. 697-717.

Reviewer 2 Report

The authors present an extremely thorough, well organized and clear description of the past, present and future state of the ways in which the study of tumor metabolism can impact cancer progression, specifically in prostate cancer. I dont have any meaningful comments to improve this work.

Author Response

Thank you so much for taking the time to provide such excellent comments to our manuscript.